# Verity plots: A novel method of visualizing reliability assessments of artificial intelligence methods in quantitative cardiovascular magnetic resonance

Thomas Hadler [1,2,3]*, Clemens Ammann[1,2,3,4], Hadil Saad[1,2,3], Leonhard Grassow[1,2,3], Philine Reisdorf[1,2,3], Steffen Lange[5], Sascha Däuber[6], Jeanette Schulz-Menger [1,2,3,4]

**1** Charité – Universitätsmedizin Berlin, Corporate Member of Freie Universität Berlin and Humboldt-Universität zu Berlin, ECRC Experimental and Clinical Research Center, Berlin, Germany, **2** Working Group on CMR, Experimental and Clinical Research Center, a Joint Cooperation Between the Max-Delbrück-Center for Molecular Medicine and the Charité – Universitätsmedizin Berlin, Berlin, Germany, **3** DZHK (German Centre for Cardiovascular Research), Berlin, Germany, **4** Department of Cardiology and Nephrology, Helios Hospital Berlin-Buch, Berlin, Germany, **5** Department of Computer Sciences, Hochschule Darmstadt - University of Applied Sciences, Darmstadt, Germany, **6** Bracco Imaging S.p.A., Milano, Italy

* thomas.hadler@charite.de

## Abstract

### Background

Artificial intelligence (AI) methods have established themselves in cardiovascular magnetic resonance (CMR) as automated quantification tools for ventricular volumes, function, and myocardial tissue characterization. Quality assurance approaches focus on measuring and controlling AI-expert differences but there is a need for tools that better communicate reliability and agreement. This study introduces the Verity plot, a novel statistical visualization that communicates the reliability of quantitative parameters (QP) with clear agreement criteria and descriptive statistics.

### Methods

Tolerance ranges for the acceptability of the bias and variance of AI-expert differences were derived from intra- and interreader evaluations. AI-expert agreement was defined by bias confidence and variance tolerance intervals being within bias and variance tolerance ranges. A reliability plot was designed to communicate this statistical test for agreement. Verity plots merge reliability plots with density and a scatter plot to illustrate AI-expert differences. Their utility was compared against Correlation, Box and Bland-Altman plots.

### Results

Bias and variance tolerance ranges were established for volume, function, and myocardial tissue characterization QPs. Verity plots provided insights into statstistcal

**Data availability statement:** All relevant numerical data are within the manuscript and its Supporting Information files. The MRI data (images and workspaces) used in this article cannot be shared publicly because they contain patient data and cannot be published for legal and privacy reasons. The MRI datasets are available from the data protection office at datenschutzbeauftragte@charite.de on reasonable request.

**Funding:** The author(s) received no specific funding for this work.

**Competing interests:** The authors have declared that no competing interests exist.

**Abbreviations:** AHA, American Heart Association; AI, Artificial Intelligence; BTR, Bias tolerance range; CI, Confidence interval; CMR, Cardiovascular magnetic resonance; TR, Tolerance range; ES, End-systole; ED, End-diastole; ESV, End-systolic volume; EDV, End-diastolic volume; EF, Ejection fraction; LoA, Limits of agreement; LV, Left ventricle; Midv, Midventricular; PM, Papillary muscle; QP, Quantitative parameter; RV, Right ventricle; SV, Stroke volume; TI, Tolerance interval; VTR, Variance tolerance range

properties, outlier detection, and parametric test assumptions, outperforming Correlation, Box and Bland-Altman plots. Additionally, they offered a framework for determining the acceptability of AI-expert bias and variance.

## Conclusion

Verity plots offer markers for bias, variance, trends and outliers, in addition to deciding AI quantification acceptability. The plots were successfully applied to various AI methods in CMR and decisively communicated AI-expert agreement.

## Introduction

In the last two decades cardiovascular magnetic resonance (CMR) has evolved from a qualitative imaging technique to an evermore quantitative field, becoming the gold-standard for volume and function quantification [1]. Furthermore, CMR offers techniques to differentiate myocardial tissue, such as parametric mapping [2]. Since quantitative parameters (QP) can be influenced by a multitude of confounders, including different readers [3], artificial intelligence (AI) methods [4], and sites [5], ensuring the reliability of QPs has increased in importance. More specifically, the recent success of AI-driven quantification has led to QP results close to experts [6], however, they also revealed unforeseen difficulties with QP evaluations failing to generalize to sites and scanners [7,8].

When AI methods are introduced in research, they are compared to experts that evaluate the QP for the same patient cohort [4,9–12]. Based on statistical properties of the differences between the expert and the AI, the AI's QP evaluations are deemed acceptable or unacceptable. More generally, determining a new evaluation method's reliability requires the statistical comparison to established evaluation methods. Reliability refers to the consistency of QP evaluations under equal conditions, including both interobserver (consistency among different observers) and intraobserver (consistency for one observer across trials) reliability. To ensure that all definitions used are understood consistently, they are explained below.

### Definitions

Reliability tests are statistical tests that decide whether two evaluation methods agree when evaluating the same data. Confidence intervals provide a range around the sample mean where the true population mean is likely to fall [13,14]. Narrow confidence intervals indicate less uncertainty and can be used to assess whether repeated evaluations are consistently close. The lesser known tolerance intervals define a range that is expected to contain a specific proportion of the population with a given level of confidence [14]. They help determine whether most evaluations will fall within an acceptable range. In CMR, a tolerance range was initially designed to cover 95% of intrareader differences [15]. This tolerance range was then used to assert equivalence when a new evaluation method's confidence interval was covered by it. Over time, these tolerance ranges were generalized for use in quality assurance

analyses within CMR and applied to evaluate the acceptability of AI-based QP evaluations [4,5,16–18]. However, they focus on the acceptability of the bias, without addressing the variance. This seems suboptimal because QP evaluation concerns individual patients, which may deviate from the bias – these deviations should be known and limited.

**Visualizations of statistical tests**

In order to communicate reliability tests and dataset characteristics more effectively, statistical plots were developed. Correlation plots [19], box plots [20], and Bland-Altman plots [21] are common visualization methods to communicate the results of statistical tests. However, these plots have limitations that may poorly communicate important evaluation method differences (Fig 1). Correlation plots illustrate correlational relationships between evaluation pairs, but present bias poorly [21,22]. Box plots present the median, and scales to many dataset comparisons, however, they gloss over evaluation pairs and may conceal trends and skewedness. Bland-Altman plots present bias and evaluation pairs, which reveals trends. However, while these conventional plots are descriptive they do not present whether two evaluation methods agree (Fig 1).

The aim of this paper is to establish a statistical plot that communicates the reliability of quantitative parameters in CMR by visualizing clear criteria for agreement in addition to descriptive statistical properties. To this end, first, a statistical test is designed to decide evaluation agreement based on bias and variance tolerance ranges. Second, a novel statistical plot is established that combines descriptive statistical properties with agreement criteria, and third, the plot's utility is demonstrated for AI assessment.

## Methods

1) Statistical Test

The reliability of a new evaluation methods is statistically tested by combining two parametric paired statistical tests, which assess the acceptability of the bias and the variance (Fig 2). Parametric tests require normally distributed data, and a sample size of $N \geq 30$.

1.1) Definition of Tolerance Range

The statistical test required two tolerance ranges – i.e., thresholds that statistical properties of the assessed QP may not exceed – one for the bias, and another for the variance. The bias tolerance range (BTR) contains 95% of **intra**observer evaluation differences (Fig 2). The intraobserver analysis reflects the minimal variability intrinsic to the images and the evaluation method. The variance tolerance range (VTR) contains 95% of **inter**observer evaluation differences. The interobserver analysis reflects the variability of the evaluation method in a clinical environment, in which multiple readers operate.

1.2) Definition of Agreement

The new evaluation method is in agreement with the original method if its bias and variance are both acceptable. For the bias the 95%-confidence interval (CI) is calculated. The bias is considered acceptable if its CI is covered by the BTR (Fig 2). For the new method's variance, a tolerance interval that covers a configurable percentage of differences is calculated. The new method's variance is considered acceptable if its tolerance interval (TI) is covered by the VTR (Fig 2).

2) Establishment of Statistical Plots: Verity Plots

Two corresponding figures were designed to communicate the results of the above statistical test. The first figure presents the results of the statistical test above for a comparison of two evaluation methods (Extended Verity Plot); the second figure presents the statistical test compactly for several comparison methods (Condensed Verity Plot) (Fig 3).

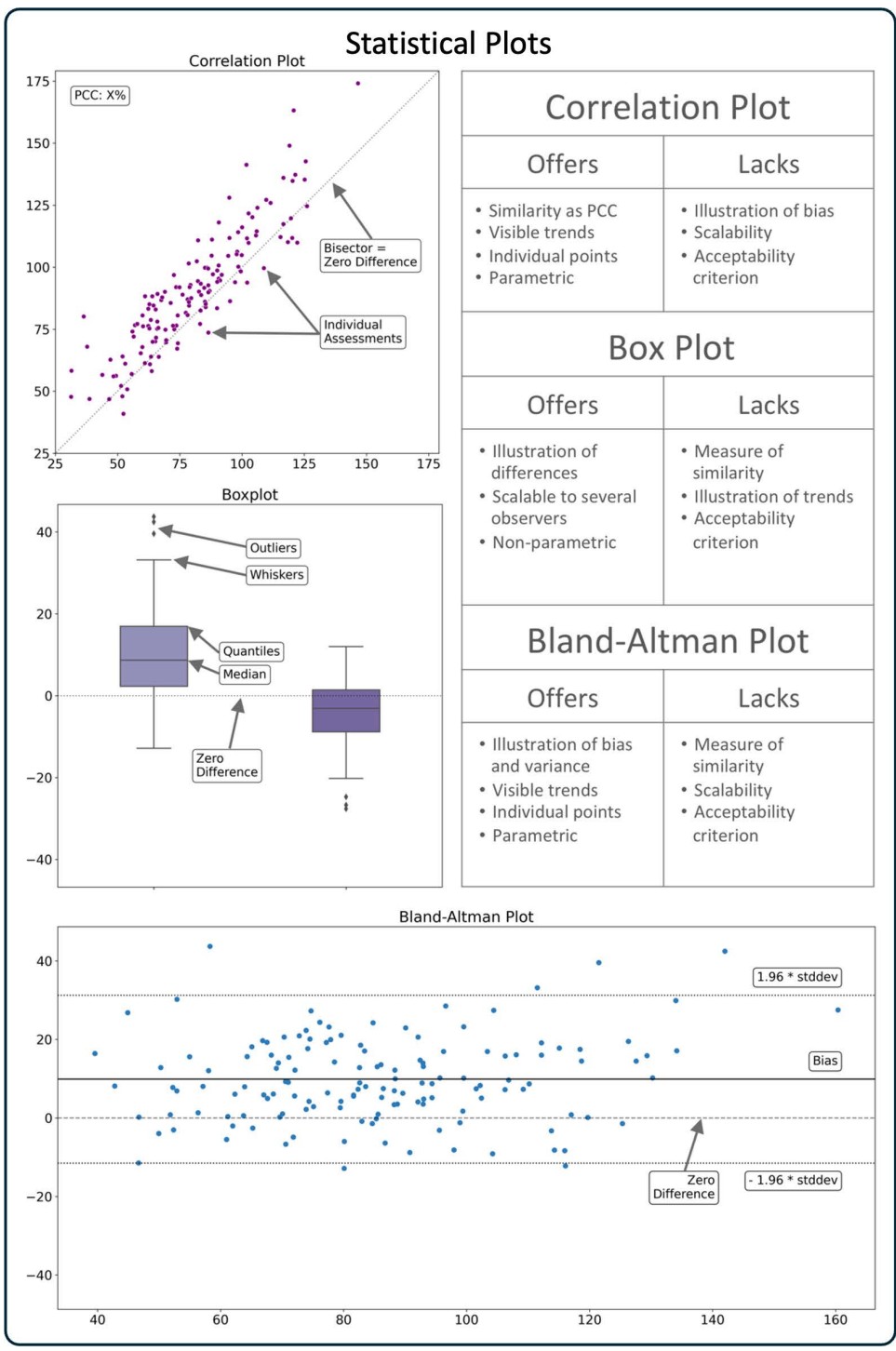

**Fig 1. Classical Statistical Plots.** Illustrative examples for correlation, box and Bland-Altman plots with capabilities and limitations listed on the right. Legend: PCC: Pearson's correlation coefficient, stddev: standard deviation.

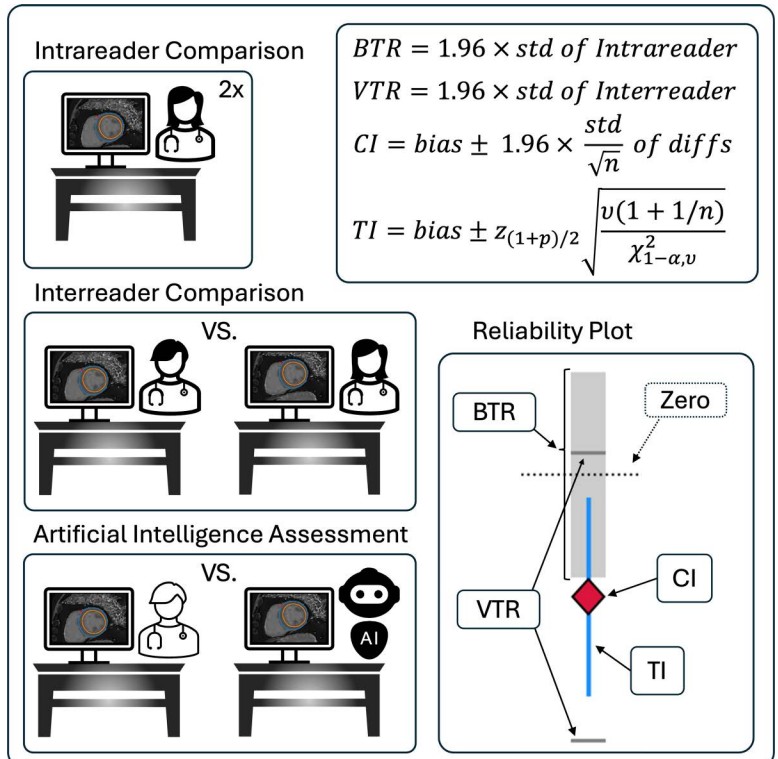

**Fig 2. Statistical Definitions and Relationships.** On the left different comparisons are shown. Intra- and interreader comparisons (same and different readers) are used to calculate tolerance ranges (BTR and VTR). The BTR/VTR contains 95% of evaluation differences for an intrareader/interreader comparison. The AI's performance is acceptable if its confidence interval (CI) is covered by the BTR, and its tolerance interval (TI) is covered by the VTR. The CI covers the bias with a confidence of 95%, the TI covers a configurable portion (i.e., $1 - \alpha$) of the differences with a confidence of 95%. These relationships are depicted in the Reliability Plot, in which the CI (red diamond) leaves the BTR (grey area), but the TI (blue lines) is covered by the VTR (dark grey lines). Legend: BTR: bias tolerance range, VTR: variance tolerance range, CI: 95%-confidence interval, TI: tolerance interval, std: standard deviation, $z_{\frac{1+p}{2}}$: critical value of normal distribution of the cumulative probability $(1+p)/2$, $\chi^2_{1-\alpha,v}$: critical value of the chi-square distribution with $9v$ degrees of freedom that is exceeded with probability $\alpha$.

## Extended verity plot

1. Presents a two-part figure with both acceptability decisions (for bias and variance) on the left, and elements pertaining to the statistical trends on the right

2. Draws the zero difference line

3. On the left the reliability decision is presented as the reliability plot (3a, 3b, 3c):

   a) Plots the bias confidence interval and color-codes the bias's acceptability

   b) Plots the variance tolerance interval and color-codes the variance's acceptability

   c) Integrates the two tolerance ranges (±BTR, Bias±VTR) as guiding figure elements to indicate reliability decisions

   d) Illustrates plausibility of normal distribution with a density plot

4. On the right:

   a) Plots the average difference of the evaluation method in comparison to the reference evaluation (bias)

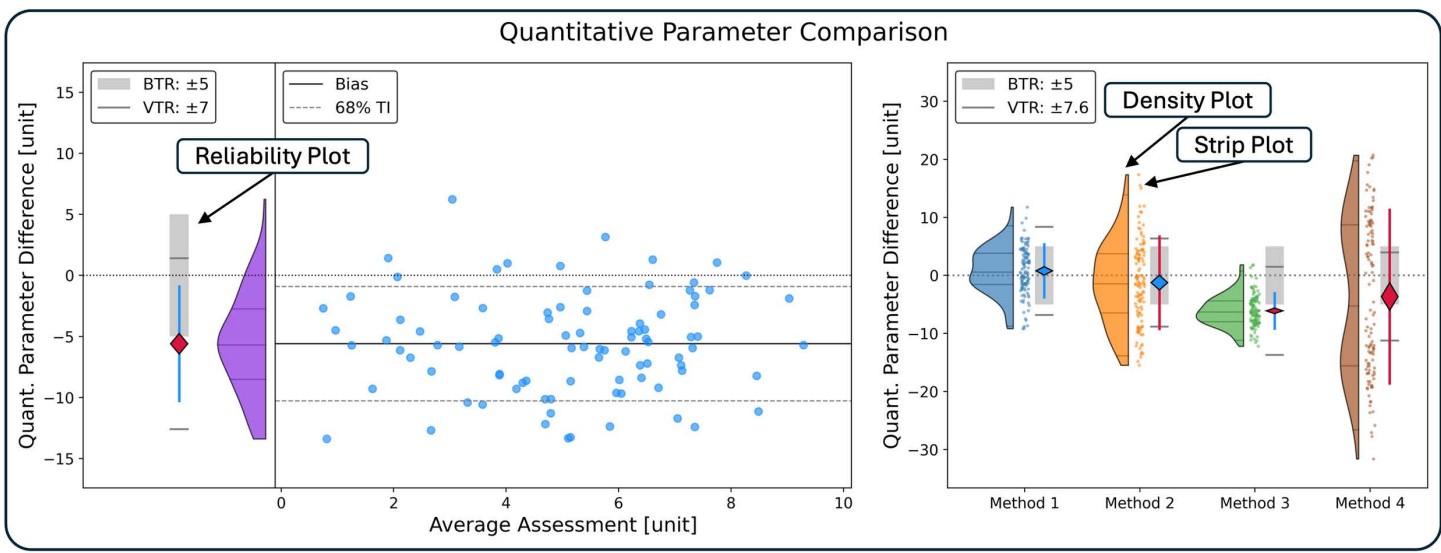

**Fig 3. Extensive and Condensed Verity Plots.** The extensive Verity plot (left) visualizes a measurement comparison. The left half shows a reliability and a density plot. The reliability plot shows the results of the statistical test as well as its central components: the bias' confidence interval, the bias tolerance range (BTR) around zero, the variance tolerance interval (TI), and the variance tolerance range (VTR) as two grey horizontal lines. Blue depicts acceptability, red unacceptability. The right half shows a Bland-Altman style difference plot, in which the bias and the tolerance interval are shown. The condensed Verity plot (right) for multiple comparisons (Method 1–4) shows a density, strip, and reliability triplet per method comparison.

b) Plots the 95%-tolerance interval limits of V% of covered differences (bias±V%-tolerance interval) of the repeated evaluation

c) Plots the paired data points as averages (or reference evaluation values) on the x-axis and differences on the y-axis

i) This visualizes potential trends and skewedness

**Condensed verity plot**

1. Draws the line for equality of evaluation (zero difference)

2. Presents multiple dataset comparisons simultaneously

   a. A density plot supports trend and skewedness detection

   b. The paired data points are plotted without x-values, collapsing vertically to present only the differences

   c. A reliability decision plot is plotted as in the first figure

3) Testing the Statistical Plots

The Verity plots are used to evaluate an AI method for cardiac image segmentation of short-axis cine data as well as an AI method for parametric mapping reference point detection and segmentation. To this end, tolerance ranges are created, several AI methods are trained, the Verity plots are assessed in diverse settings and compared to classical statistical plots. Finally, customization options are presented for the plots.

### 3.1) Creation of Tolerance Ranges

For 30 CMR studies, volumes and function as well as T1 and T2 parametric values were evaluated twice by one expert reader, and once by a second expert. The volume and function QPs were the left and right ventricular (LV, RV) end-systolic, end-diastolic and stroke volumes (ESV, EDV, SV), ejection fraction (EF), LV myocardial mass (LVM) and ED papillary muscle mass (PM). The parametric mapping values were the midventricular average value (T1 PRE Midv Avg, T2 Midv Avg) as well as six segmental values based on dividing the midventricular slice into segments with a reference point according to the American Heart Association model (T1 PRE AHA_(n), T2 AHA_(n) for n in 7–12). For each QP the intra- and interobserver differences were calculated. For each parameter ±1.96 standard deviations of intraobserver and interobserver differences defined the BTR and the VTR.

### 3.2) Assessing Visualization Capabilities of Verity Plots

In order to assess the capabilities of Verity plots, we trained multiple AI methods, used Verity plots to evaluate the acceptability of their QPs, and compared the visualizations to traditional alternatives.

**Training of artificial intelligence methods**

For short-axis cine, we trained a U-Net [23] on 150 cases. The 150 cases were annotated by an expert in ES and ED with LV endocardial, myocardial and papillary muscle contours, as well as RV endocardial contours. We created four AI methods by varying the rasterizations for the different cardiac structures during training that increased the LV myocardium and papillary muscles, while decreasing the LV and RV endocardial structures. For parametric mapping methods, we trained three U-Nets on 224 cases for a cascaded annotation AI, in which a first U-Net detects the LV, which is enlarged for subpixel-accuracy and annotated by two further U-Nets [16]. The first contours the myocardium, and the second detects the reference point, at which the RV connects to the LV myocardium. During evaluation several security rims are used for the AI methods (−5%, 0%, 5%, 10%) to provide multiple QP results. For an in-depth description of the employed U-Net architecture and training procedure see the supplemental material (S1_File.pdf).

**Testing and demonstrating verity plots**

To test the utility of Verity plots, the short-axis cine AI methods were evaluated on 318 cases; the parametric mapping AI methods were evaluated on 59 cases from the same dataset. Verity plots were presented for a selected subset of QPs; all plots for all QPs are available in the appendix. Density plots within the Verity plots were used to evaluate whether a normal distribution may be assumed by visually assessing symmetry, a single peak, and quickly decreasing tails. The visualization capabilities of Verity plots were compared to traditional alternative plots: correlation plots, box plots and Bland-Altman plots. The results of this comparison are summarized in a table.

### 3.3) Customization Options

While Verity plots were designed to be extensive in their presentation, they may be reduced for simple comparison environments when it is desirable to limit visual attention to certain statistical tests or properties of the distribution. To this end, Verity plots are customizable, including options for horizontal presentation and condensing the plots by merging or removing visual elements.

More specifically, Verity plots allow for turning certain elements on and off (in- or excluding: reliability tests, density plots), in- or excluding percentiles in the density plot, moving the legend locations, and merging the density and test elements. The condensed Verity plots allow for vertical and horizontal presentation, in- or excluding reliability decisions, density plots and strip plots, and percentiles within the density plot. Both plots allow for choosing the colours of the density plot faces and points.

## Ethics declarations and data access

All numerical datasets required to reproduce the results are available in the supplemental material. The use of patient data for this study was approved by the local ethics committee of the Charité Universitätsmedizin Berlin as retrospective study (study ID: EA 1253 21). The MRI datasets cannot be shared due to institutional rules but may be available on special request. For research purposes, datasets were accessed from the 07.04.2024–01.12.2024. Datasets were anonymized such that individuals could not be identified.

## Results

1) Creation of Tolerance Ranges

The bias and variance tolerance ranges for volume, function and mapping QPs are presented in Table 1.

2) Testing and Demonstrating Verity Plots

### Utility of density function for normal distribution assessment

Initial evaluation of the short-axis cine cases revealed poor AI generalization to scanners different from those in the training dataset: the AI method failed to segment cases obtained from external scanners. Verity plots display these failures as systematic outliers that violate the normal distribution (Fig 4, top). The AI method was reassessed on the internal cases of the dataset, from which 35 externally scanned cases were removed (Fig 4, bottom).

### Plot comparison: Verity, correlation, box and Bland-Altman plots

Verity Plots were used to evaluate all QPs listed in Table 1 (for all AI methods described in Methods); the illustrations for all AI methods are available in the appendix (S1_Fig, S2_Fig, S3_Fig, S4_Fig). Three illustrative QP comparisons (T2 midventricular average, RV EDV, RV EF) are shown in Fig 5 for extended Verity, correlation, box, and Bland-Altman plots.

**Table 1. Bias and Variance Tolerance Ranges for Function, Volume and Parametric Mapping Parameters.**

| Parameter | BTR | VTR | Parameter | BTR | VTR |
|---|---|---|---|---|---|
| LV ESV [ml] | 8.2 | 16.6 | T1 Midv Avg | 13.5 | 16.8 |
| LV EDV [ml] | 12.2 | 26.1 | T1 AHA_1 | 17.6 | 32.5 |
| LV SV [ml] | 13.6 | 16.9 | T1 AHA_2 | 20.8 | 26.9 |
| LV EF [%] | 7.0 | 7.8 | T1 AHA_3 | 23.1 | 25.6 |
| LV MASS [g] | 19.5 | 41.9 | T1 AHA_4 | 22.6 | 37.6 |
| LV ESPM [g] | 3.7 | 3.8 | T1 AHA_5 | 28.5 | 33.0 |
| LV EDPM [g] | 3.4 | 3.7 | T1 AHA_6 | 32.3 | 37.4 |
| RV ESV [ml] | 10.4 | 15.4 | T2 Midv Avg | 1.0 | 1.4 |
| RV EDV [ml] | 12.1 | 28.5 | T2 AHA_1 | 1.6 | 1.9 |
| RV SV [ml] | 11.4 | 20.8 | T2 AHA_2 | 2.2 | 2.3 |
| RV EF [%] | 5.6 | 6.1 | T2 AHA_3 | 1.2 | 1.4 |
| LV ES [#] | 1.3 | 1.5 | T2 AHA_4 | 1.3 | 1.7 |
| LV ED [#] | 0.9 | 1.2 | T2 AHA_5 | 2.0 | 2.0 |
| RV ES [#] | 1.6 | 4.4 | T2 AHA_6 | 1.3 | 1.3 |
| RV ED [#] | 1.2 | 1.3 | | | |

BTR, Bias Tolerance Range; VTR, Variance Tolerance Range; LV, left ventricle; RV, right ventricle; ES, end-systolic; ED, end-diastolic; ESV, end-systolic volume; EDV, end-diastolic volume; SV, stroke-volume; EF, ejection fraction; PM, papillary muscle mass; Midv Avg, pixel value average of midventricular slice; AHA, American heart association model

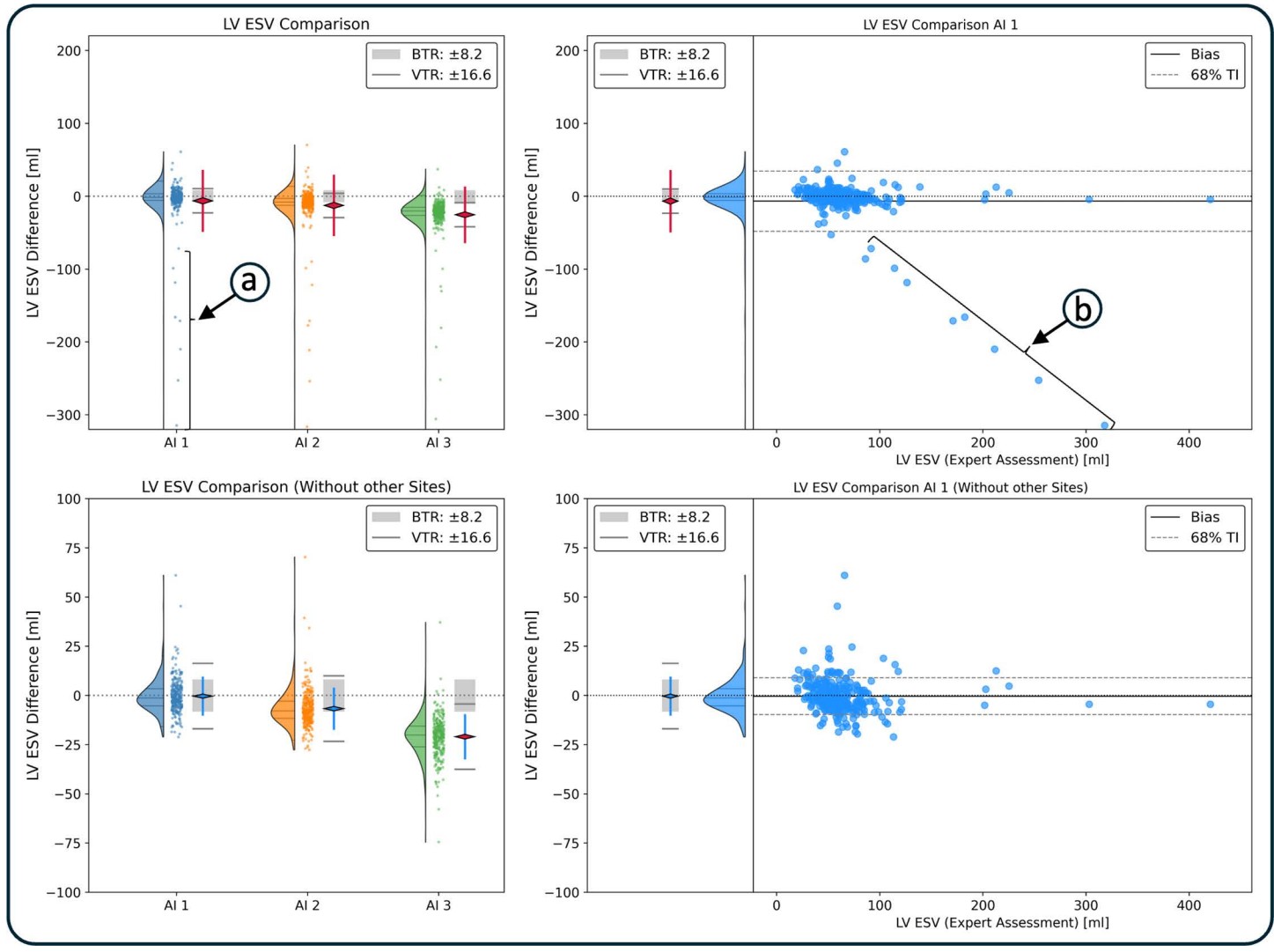

**Fig 4. Utility of Density Function for Outlier Detection in Verity Plots.** The top two figures depict three AI-expert comparisons in condensed Verity plots (left), and of the first AI to the expert in the extended Verity plot (right). Density and point plots reveal systematic outliers **(a, b)**. In the right plot the linear trend of the outliers indicate a systematic failure **(b)**. The systematic outliers break the normal distribution and invalidate the test. These segmentation failures were removed to produce the two plots below. The data are normally distributed and the first two AI methods have acceptable biases and variances. Legend: BTR: bias tolerance range, VTR: variance tolerance range, TI: tolerance interval.

The extended Verity plots supported distinctions between acceptability, problematic outliers and assessing the relevance of trends. Correlation plots (column 2 in Fig 5) presented correlations clearly, the variability and the negative trends were difficult to see for RV EDV and RV EF, and biases could not be estimated well. Box plots (column 3 in Fig 5) communicated median differences and the position of the bulk of data in relation to the zero differences line. They gave no visual indications to expected parameter ranges since the y-axis exclusively communicated differences. The trends in RV EDV and RV EF were not visible in the box plots. Bland-Altman plots (column 4 in Fig 5) communicated mean differences, and the lines of agreement visualized the limits to the bulk of data well. Trends were discernible for RV EDV and RV EF. None of the traditional plots communicated whether the differences were acceptable. The analysis is systematically summarized in Table 2.

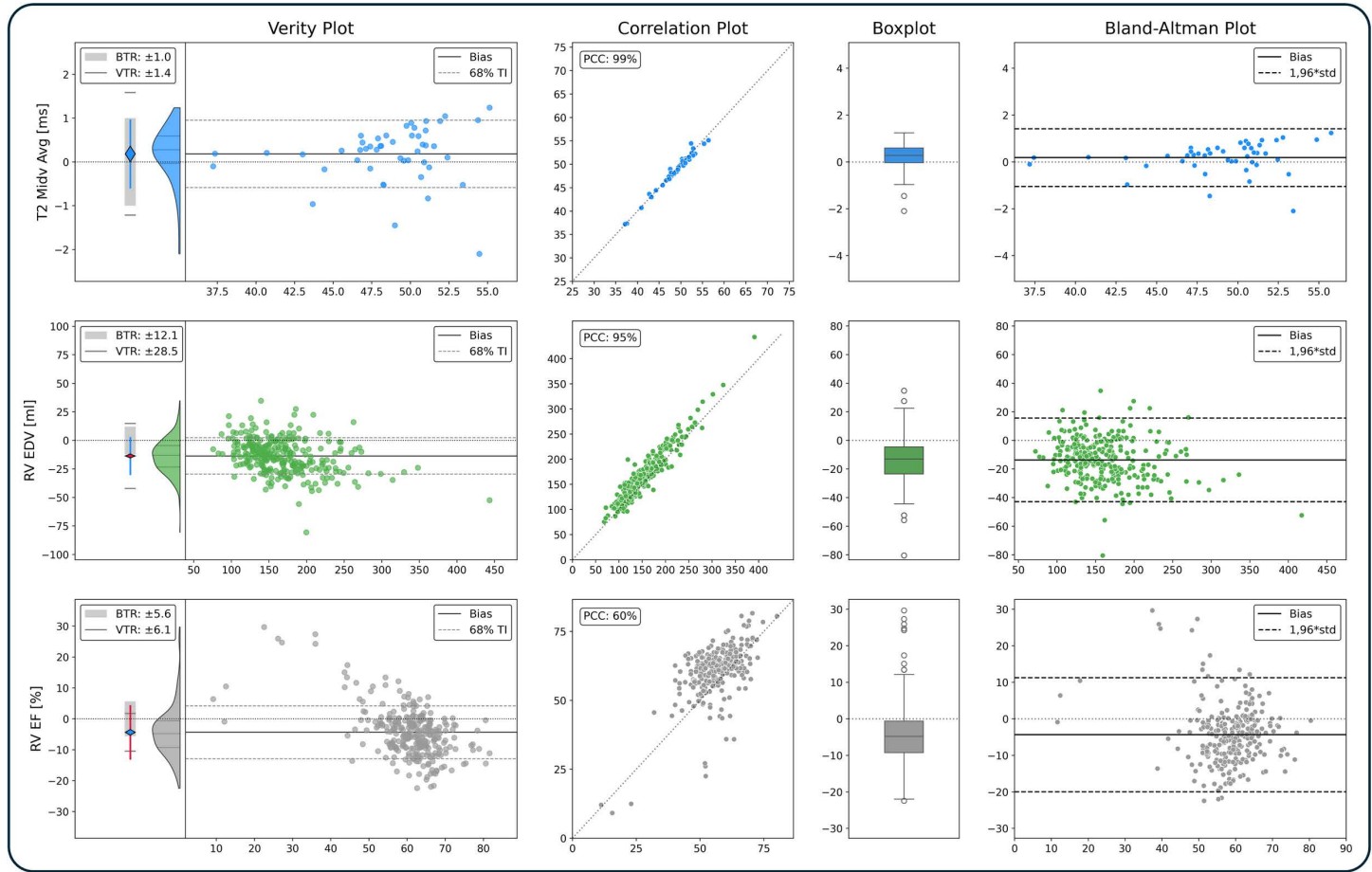

**Fig 5. Plot Comparisons: Verity, Correlation, Box and Bland-Altman Plots.** Three quantitative parameters: T1 Midventricular Average, RV EDV, and RV EF (rows 1–3) illustrate differences in expressiveness between Verity, correlation, box and Bland-Altman plots (columns 1–4). Verity plots depict the acceptability of bias and variance clearly, with negative trends visibly accessible in RV EDV and RV EF. Correlation plots show correlations and a zero difference line. PCCs are visible, biases and trends are difficult to see. Box plots offer the median, the spread (50% of differences) as the box's extent, and 98% of differences as whiskers. The data spread is visible, but trends and outliers are concealed. Bland-Altman plots show the bias, and 1.96*standard deviations as agreement limits. They show the bias and variance of data, as well as trends and outliers. Legend: BTR: bias tolerance range, VTR: variance tolerance range, TI: tolerance interval, PCC: Pearson's correlation coefficient, Midv.: midventricular, RV: right ventricle, EF: ejection fraction, EDV: end-diastolic volume.

## Customization options

Verity plots can be customized to simplify or augment information for users. In Fig 6 customizations are applied on randomly generated data. Plotted horizontally (Fig 6A) they increase the focus on how method-method differences distribute around zero. For small sample sizes the density plot may be omitted for clarity (Fig 6B). Legend positioning was changed between A and B. The third Verity plot (Fig 6C) merges the density plot with the reliability decision plot. Face and point colours were also changed. In Fig 6D the reliability plots were merged with the density plots, for which percentiles were removed. The density plot can be replaced by a box plot (Fig 6E), which may be useful when a normal distribution is not present, but a quantitative description remains useful. The final subfigure (Fig 6F) removed the reliability plot while keeping the bias confidence rhombus and variance intervals. It further switches colours between the density plot and the points and moves the density plot to the left.

**Table 2. Comparison of Reliability Visualization Methods.**

| Method | Bias | Bulk | Trend | Outlier | Agreement | Scalable |
|---|---|---|---|---|---|---|
| **Correlation Plot** | × – does not directly show bias | × – shows overall distribution but not limits for proportion | × – trends not reliably visible | ✓ – shows outliers in relation to overall distribution | × – not available | × – uses x- and y-axis |
| **Box plot** | ✓ – shows the median | ✓ – shows quantiles of the data distribution | × – cannot track systematic trends | Partially – shows single outliers | × – not available | ✓ – only y-axis |
| **Bland-Altman Plot** | ✓ – shows the mean difference | ✓ – limits of agreement contain 95% of data | ✓ – can reveal trends across x-axis | ✓ – outliers visible as outside limits of agreement | × – not available | × – uses x- and y-axis |
| **Extended Verity Plot** | ✓ – shows the mean difference + confidence interval | ✓ – bulk of data inside tolerance intervals | ✓ – can reveal trends across x-axis | ✓ – outliers leave the variance tolerance range | ✓ – shows agreement as reliability plot | × – uses x- and y-axis |
| **Condensed Verity Plot** | ✓ – shows the mean difference + confidence interval | ✓ – bulk of data inside tolerance intervals | × – only uses y-axis, x-axis collapsed | ✓ – outliers leave the variance tolerance range | ✓ – shows agreement as reliability plot | ✓ – only y-axis |

A qualitative comparison of Verity Plots and traditional reliability visualization methods (i.e., correlation, box, and Bland-Altman plots) across key assessment criteria. Verity plots offer advantages in detecting trends, outliers, and the concentration of data, while traditional methods have limitations in communicating distribution characteristics or agreement.

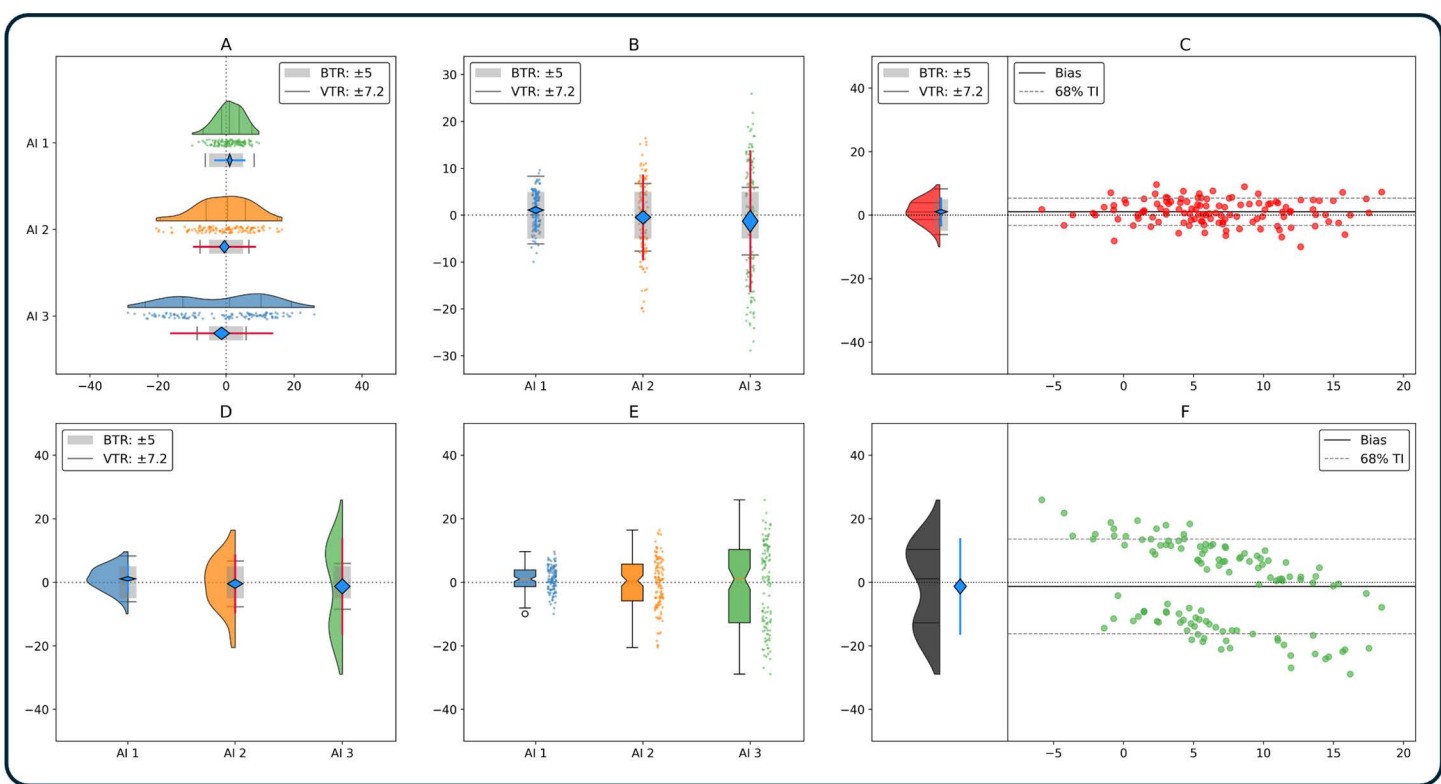

**Fig 6. Customization Options of Verity Plots.** Condensed Verity plots can be plotted horizontally (A) to focus attention on the bias and variability around the vertical zero difference line. In (B) the density plots were removed to focus on the statistical test; the spread of differences is presented with the plotted points. In (C) the Verity plot's density plot was merged with the reliability plot, colours were changed to red. In (D) the points were removed, and the density plot was merged with the reliability test. In (E) the density plots were replaced with box plots, and the reliability tests were removed. In (F) the density plot was moved entirely to the left, and colour coding was chosen separately. Legend: BTR: bias tolerance range, VTR: variance tolerance range, TI: tolerance interval, AI: artificial intelligence.

## Discussion

Verity plots, were successfully designed, implemented, and tested to show that communicating the reliability of quantitative parameters and visualizing clear agreement criteria is possible and feasible. To this end, bias and variance tolerance ranges for volume, function and parametric mapping techniques in CMR were calculated. These tolerance ranges were applied as decision criteria in a statistical test to determine whether two evaluation methods agree on evaluating a quantitative parameter. Finally, this statistical test was effectively integrated into novel statistical plots, called Verity plots, that convey the results of the statistical test as well as statistical properties. We demonstrated the Verity plots by evaluating the reliability of QPs generated by various AI methods, comparing them to expert evaluations. Verity plots outperform traditional statistical visualizations by allowing to assess the reliability of AI-generated QP evaluations, while also enabling users to identify outliers.

### Tolerance ranges

The BTR and VTR reflect limits of reproducibility in clinical research that capture key confounding influences through intra- and interreader evaluations. Intrareader differences, by definition, compare an expert to themselves, while interreader evaluations highlight differences between two experts. Intra- and interobserver analyses thus isolate reader-specific contributions to annotation and QP differences. The persistence of variations in intrareader analyses suggests that some of these differences arise from the inherent challenges of image interpretation. Interobserver analyses may introduce additional confounders, such as differing interpretation approaches, further contributing to larger variability.

The VTR is consistently one to two times larger than the BTR in Table 1. Due to the nature of intra- and interreader differences this is reasonable, as the confounding factors that cause the variability in the BTR are also present in the VTR, while the VTR additionally contains reader-specific idiosyncrasies. However, the QPs differ in how strongly they are affected by other readers. For volume QPs the VTRs are about twice as large, stroke volumes are close to 1.5, and ejection fractions are 1.1 times larger than BTRs. This makes sense since consistent contouring differences between readers impact end-systolic and diastolic volumes equally and cancel out in the function QPs. Interestingly, mapping parameters were quite resistant to interreader confounders (factor 1.25–1.4). Possibly because myocardial contour thickness differences do not add up, and may be compensated for by the 5% safety-rim.

### Statistical plots for quantitative parameter assessments

The classical correlation, box and Bland-Altman plots were used as plot-benchmarks since they provide timeless statistical visualization tools. Correlation plots are nearly 200 years old [19,24,25], and, while box and Bland-Altman plots are more recent (1977 and 1986), they are omnipresent in all quantitative fields [20,21]. Bland's and Altman's goal was to provide a method that accounted for bias in assessing agreement, rather than relying solely on correlation as a sufficient measure to define agreement [22]. However, while they highlight differences, they lack the means to convey whether the level of difference between methods is acceptable. Verity plots build on the Bland-Altman approach by incorporating the difference plot but replacing the 95% limits of agreement (LoA) with tolerance intervals. This adaptation is necessary for compatibility with the reliability tests, as comparing the LoA – which is the 95% confidence interval for differences – with the VTR would be inappropriate since the VTR itself represents the 95% confidence interval of another reader's differences.

Modifications to the Bland-Altman plot have been proposed, such as enhancing the LoA with confidence intervals [26–28], which show the likely upper and lower bounds of the LoA. However, these changes are often not well-justified because Bland-Altman plots are not designed to assess the precision of the LoA itself. Instead, they aim to use the LoA to ensure that a sufficiently large percentage of differences fall within the limits. For this reason, confidence intervals around the LoA do not add meaningful information, they rather complicate the plot and add a misleading indicator. In contrast,

tolerance intervals are the correct statistical choice to replace the LoA lines: they provide a better limit for the variance while accounting for uncertainty.

In recent decades modern plots such as violin plots [29], raincloud plots [30], bean plots [31], and swarm plots [32] have emerged as potent visualization tools in exploratory data analysis. Violin plots are two-sided kernel-density plots (as are the density plot in Verity plots) merged with the quantile information of a box plot [29,33,34]. Bean plots are similar to violin plots but include the individual data points ("beans") for visualization of small datasets [31]. Condensed Verity plots provide a similar unification of information. Raincloud plots are similar to Verity plots in the regard that they merge density with strip plots. However, they opt for integrated box plots rather than the reliability plots making raincloud plots nonparametric and descriptive. Swarm plots display individual data points next to each other to avoid overlapping [32]. For medium sized datasets this allows for a visual estimation of the density. However, none of these plots integration the decision criteria that were central to the design of Verity plots.

### AI development supported by Verity plots

The central issue of developing reliable AI for automatic QP generation in CMR is the vast space of hyperparameters. This includes numerous tuneable parameters that define an AI method, such as data augmentation strategies, neural network architecture choices, as well as the integration of geometrical assumptions into the AI pipeline. For instance, cardiac structure segmentation networks may be based on diverse architectures, such as the Fully Convolutional Network [11,35], the U-Net [23], or several variations (e.g., Attention U-Net [36], Trans-U-Net [37], etc.). These architectures may further vary layer types, channel depths, convolution sizes, normalization methods, and activation functions. Similarly extensive lists of hyperparameters apply to data augmentations and geometrical assumptions.

Previously, AI optimization for segmentation tasks has focused on manually designing architectures and fine-tuning hyperparameters [38]. When automated, hyperparameter optimization often targeted a small set of parameters using optimization-libraries [39,40]. More recently, however, the field has shifted towards training populations of AI models that explore the hyperparameter space, using intermittent evaluations to identify promising configurations [41,42]. Condensed Verity plots, with their integrated reliability tests, provide a visually intuitive way to assess the performance of AI populations as they optimize and refine their hyperparameters.

### Limitations

Verity plots were exclusively tested on quantitative parameters in CMR and could be difficult to extend to or apply in other quantitative areas. Verity plots, like all visualization methods, rely on human interpretation, which may introduce variability in assessment across different users. Future studies should evaluate their usability across a broader range of clinical and technical settings. As a parametric test, the agreement test communicates whether two QP assessment methods agree, assuming that their differences are normally distributed and a sample size larger than 30 is provided. Non-parametric confidence and tolerance intervals for a non-parametric agreement test are possible but would require larger sample sizes, which are often infeasible for QPs based on medical imaging.

### Conclusion

We introduced bias and variance tolerance ranges to assess the reliability of new evaluation methods in CMR with a custom statistical test. We integrated a visualization of this test into a novel statistical plot, the Verity plot, which clearly conveyed the reliability of new evaluation methods when compared to expert evaluations. These plots include indicators for the bias, variance, as well as trends and outliers, and most importantly Verity plots include decision criteria for parameter acceptability. They were helpful at quickly conveying important features of AI-expert differences across three different imaging techniques. Verity plots are expressive and decisive plots that we expect to guide the development of AI methods in CMR for years to come.

## Supporting Information

**S1_Fig.  Short-Axis Cine Clinical Parameter Differences for Artificial Intelligence Methods.** For all four AI methods, and for cases from all available sites, Condensed Verity Plots are visualized to communicate AI agreement with expert. (PNG)

**S2_Fig.  Short-Axis Cine Clinical Parameter Differences for Artificial Intelligence Methods – Single Site.** For all four AI methods, and for cases from one site, Condensed Verity Plots are visualized to communicate AI agreement with expert. (PNG)

**S3_Fig.  T1 Mapping Clinical Parameter Differences for Artificial Intelligence Methods.** For all four AI methods, and for cases from all available sites, Condensed Verity Plots are visualized to communicate AI agreement with expert. (PNG)

**S4_Fig.  T2 Mapping Clinical Parameter Differences for Artificial Intelligence Methods.** For all four AI methods, and for cases from all available sites, Condensed Verity Plots are visualized to communicate AI agreement with expert. (PNG)

**S1_File.**   AI Architecture and Training Pipeline. Describes the U-Net architecture details employed for training, as well as details of the training procedures, including image normalization and augmentation strategies. (PDF)

**DataArchive.zip.**   Supporting Information - Tabular Data Includes all tabular data used to produce the quantitative analyses, figures and tables in this manuscript. (ZIP)

## Acknowledgments

We wish to thank the members of the working group on CMR for input at different steps.

## Author contributions

**Conceptualization:** Thomas Hadler, Clemens Ammann, Hadil Saad, Philine Reisdorf, Steffen Lange, Sascha Däuber, Jeanette Schulz-Menger.

**Data curation:** Thomas Hadler, Hadil Saad, Leonhard Grassow.

**Formal analysis:** Thomas Hadler.

**Methodology:** Thomas Hadler, Clemens Ammann, Hadil Saad, Steffen Lange, Jeanette Schulz-Menger.

**Project administration:** Thomas Hadler.

**Software:** Thomas Hadler, Clemens Ammann.

**Supervision:** Jeanette Schulz-Menger.

**Visualization:** Thomas Hadler.

**Writing – original draft:** Thomas Hadler, Jeanette Schulz-Menger.

**Writing – review & editing:** Thomas Hadler, Clemens Ammann, Hadil Saad, Leonhard Grassow, Philine Reisdorf, Steffen Lange, Sascha Däuber, Jeanette Schulz-Menger.

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
