## [Decision Letter · Decision Letter 0]

23 Feb 2025

PONE-D-25-01126Verity Plots – A Novel Method of Visualizing Reliability Assessments of Artificial Intelligence Methods in Quantitative Cardiovascular Magnetic ResonancePLOS ONE

Dear Dr. Hadler

Thank you for submitting your manuscript to PLOS ONE. After careful consideration, we feel that it has merit but does not fully meet PLOS ONE’s publication criteria as it currently stands. Therefore, we invite you to submit a revised version of the manuscript that addresses the points raised during the review process.

Please submit your revised manuscript by Apr 09 2025 11:59PM. If you will need more time than this to complete your revisions, please reply to this message or contact the journal office at plosone@plos.org . Please include the following items when submitting your revised manuscript:

We look forward to receiving your revised manuscript.

Kind regards,

Haipeng Liu

Academic Editor

PLOS ONE

3. Please include a copy of Table 2 which you refer to in your text on page 14.

Additional Editor Comments (if provided):

Reviewers' comments:

Reviewer's Responses to Questions

**Comments to the Author**

1. Is the manuscript technically sound, and do the data support the conclusions?

Reviewer #1: Yes

Reviewer #2: Partly

2. Has the statistical analysis been performed appropriately and rigorously? 

Reviewer #1: Yes

Reviewer #2: N/A

3. Have the authors made all data underlying the findings in their manuscript fully available?

Reviewer #1: Yes

Reviewer #2: Yes

4. Is the manuscript presented in an intelligible fashion and written in standard English?

Reviewer #1: Yes

Reviewer #2: Yes

5. Review Comments to the Author

Reviewer #1: Thank you for submitting your manuscript "Verity Plots – A Novel Method of Visualizing Reliability Assessments of Artificial Intelligence Methods in Quantitative Cardiovascular Magnetic Resonance." I have thoroughly reviewed your work and find it to be an innovative contribution to the field of AI in cardiovascular magnetic resonance imaging. The introduction of Verity Plots as a novel visualization method for assessing AI reliability is both timely and valuable

Reviewer #2: The aim of this paper” Verity Plots – A Novel Method of Visualizing Reliability Assessments of Artificial Intelligence Methods in Quantitative Cardiovascular Magnetic Resonance” is to establish a statistical plot that communicates the reliability of quantitative parameters in CMR by visualizing clear criteria for agreement in addition to descriptive statistical properties. To this end, first, a statistical test is designed to decide evaluation agreement based on bias and variance tolerance ranges. Second, a novel statistical plot is established that combines descriptive statistical properties with agreement criteria, and third, the plot’s utility is demonstrated for AI assessment.

Good work but still need ...

all comments with the editor

6. PLOS authors have the option to publish the peer review history of their article (what does this mean? ). If published, this will include your full peer review and any attached files.

**Do you want your identity to be public for this peer review?** For information about this choice, including consent withdrawal, please see our Privacy Policy .

Reviewer #1: **Yes: ** Luqman Adewale Abass

Reviewer #2: No

---

## [Author Response · Author response to Decision Letter 1]

3 Mar 2025

We thank the editor and the reviewers for their thoughtful comments. We believe that the manuscript has benefited greatly from these remarks, and are glad to offer our full response as the uploaded file: "Response to Reviewers.docx" as well as the updated manuscript.

---

## [Decision Letter · Decision Letter 1]

8 Apr 2025

Verity Plots – A Novel Method of Visualizing Reliability Assessments of Artificial Intelligence Methods in Quantitative Cardiovascular Magnetic Resonance

PONE-D-25-01126R1

Dear Dr. Hadler,

We’re pleased to inform you that your manuscript has been judged scientifically suitable for publication and will be formally accepted for publication once it meets all outstanding technical requirements.

Kind regards,

Haipeng Liu

Academic Editor

PLOS ONE

Additional Editor Comments (optional):

Reviewers' comments:

Reviewer's Responses to Questions

**Comments to the Author**

1. If the authors have adequately addressed your comments raised in a previous round of review and you feel that this manuscript is now acceptable for publication, you may indicate that here to bypass the “Comments to the Author” section, enter your conflict of interest statement in the “Confidential to Editor” section, and submit your "Accept" recommendation.

Reviewer #2: All comments have been addressed

2. Is the manuscript technically sound, and do the data support the conclusions?

Reviewer #2: Yes

3. Has the statistical analysis been performed appropriately and rigorously? 

Reviewer #2: Yes

4. Have the authors made all data underlying the findings in their manuscript fully available?

Reviewer #2: Yes

5. Is the manuscript presented in an intelligible fashion and written in standard English?

Reviewer #2: Yes

6. Review Comments to the Author

Reviewer #2: good work keep up

all comments are submitted , no extra comments, excellent efforts , the article worth to be published

7. PLOS authors have the option to publish the peer review history of their article (what does this mean? ). If published, this will include your full peer review and any attached files.

**Do you want your identity to be public for this peer review?** For information about this choice, including consent withdrawal, please see our Privacy Policy .

Reviewer #2: No

---

## [Editor Report · Acceptance letter]

PONE-D-25-01126R1

PLOS ONE

Dear Dr. Hadler,

I'm pleased to inform you that your manuscript has been deemed suitable for publication in PLOS ONE. Congratulations! Your manuscript is now being handed over to our production team.

Kind regards,

on behalf of

Dr. Haipeng Liu

Academic Editor

PLOS ONE